# Bioactivity and Development of Small Non-Platinum Metal-Based Chemotherapeutics

**DOI:** 10.3390/pharmaceutics14050954

**Published:** 2022-04-28

**Authors:** Maria Grazia Ferraro, Marialuisa Piccolo, Gabriella Misso, Rita Santamaria, Carlo Irace

**Affiliations:** 1BioChemLab, Department of Pharmacy, School of Medicine and Surgery, University of Naples “Federico II”, Via D. Montesano 49, 80131 Naples, Italy; mariagrazia.ferraro@unina.it (M.G.F.); marialuisa.piccolo@unina.it (M.P.); rita.santamaria@unina.it (R.S.); 2Department of Precision Medicine, School of Medicine and Surgery, University of Campania “Luigi Vanvitelli”, 80138 Naples, Italy

**Keywords:** metal-based chemotherapeutics, chemotherapy, chemoresistance, ruthenium-based drugs, palladium-based drugs, rhodium-based drugs, iridium-based drugs, gold-based drugs, platinum-based drugs

## Abstract

Countless expectations converge in the multidisciplinary endeavour for the search and development of effective and safe drugs in fighting cancer. Although they still embody a minority of the pharmacological agents currently in clinical use, metal-based complexes have great yet unexplored potential, which probably hides forthcoming anticancer drugs. Following the historical success of cisplatin and congeners, but also taking advantage of conventional chemotherapy limitations that emerged with applications in the clinic, the design and development of non-platinum metal-based chemotherapeutics, either as drugs or prodrugs, represents a rapidly evolving field wherein candidate compounds can be fine-tuned to access interactions with druggable biological targets. Moving in this direction, over the last few decades platinum family metals, e.g., ruthenium and palladium, have been largely proposed. Indeed, transition metals and molecular platforms where they originate are endowed with unique chemical and biological features based on, but not limited to, redox activity and coordination geometries, as well as ligand selection (including their inherent reactivity and bioactivity). Herein, current applications and progress in metal-based chemoth are reviewed. Converging on the recent literature, new attractive chemotherapeutics based on transition metals other than platinum—and their bioactivity and mechanisms of action—are examined and discussed. A special focus is committed to anticancer agents based on ruthenium, palladium, rhodium, and iridium, but also to gold derivatives, for which more experimental data are nowadays available. Next to platinum-based agents, ruthenium-based candidate drugs were the first to reach the stage of clinical evaluation in humans, opening new scenarios for the development of alternative chemotherapeutic options to treat cancer.

## 1. An Outline on the Classical Metallochemotherapeutics: Cisplatin and Congeners

Looking at the landscape of metal-based chemotherapy, the scene has thus far been dominated by platinum-based drugs such as cisplatin (*c*DDP) and congeners [1,2]. This story originated in 1965, when Rosenberg made the unforeseen discovery that a metal-based compound, cis-diamminedichloridoplatinum (II), later called cisplatin, retained biological activity. This was an outstanding breakthrough in the history of chemotherapy. Entered in clinical trials way back in 1971 and approved in 1978, first by the Food and Drug Administration for the treatment of testicular and ovarian cancer and then by other European countries, cisplatin is still considered as a reference drug and included in the WHO list of essential medicines [3,4]. This small platinum-based molecule—so different from current therapeutics developed by modern researchers—is yet a largely used drug in the clinical treatment of many solid tumours. Indeed, its use has been so extensive that it earned it the nickname of the “penicillin of cancer” [5,6]. As such, cisplatin inspired the way for the development of numerous Pt(II)-based metal coordination complex derivatives or analogues which have attracted extensive interest as novel prospective anticancer agents [2,7,8]. In the wake of its success, first carboplatin and then oxaliplatin were developed and entered clinical use starting in 1980 [9,10,11]. Moreover, nedaplatin, lobaplatin, and heptaplatin were subsequently fabricated as second-generation cisplatin analogues and approved for clinical use only in specific countries [5,12,13]. A timeline concerning the development of the main Pt(II)-based anticancer drugs and their subsequent approval in the clinic is depicted in Figure 1. Altogether, dozens of Pt(II) complexes have reached clinical studies. Thus, at the turn of the 21st century, platinum-based drugs are being used in the treatment of approximately 50–70% of cancers [2,5,8,14,15,16]. Overall, cisplatin and carboplatin can certainly be considered the most successful platinum-based metallochemotherapeutics in history, having demonstrated efficacy in various human cancers, including lung, ovarian, testicular, bladder, neck, and pancreatic cancer, as well as effectiveness against various types of cancer cells, i.e., carcinomas, germ cell tumours, lymphomas, and sarcomas [9,10,17,18].

The primary target of cisplatin is chromosomal DNA. This interaction induces a cascade of cytotoxic effects, starting from the damage of the genetic material and ending with the activation of apoptosis and cell death [19]. Nevertheless, the molecular mechanisms by which DNA damage results in the activation of apoptotic pathways are still under investigation [20]. Clarification of such a complex process would be of extraordinary importance in the understanding of apoptosis evasion and cell survival, as well as in clinical prediction of tumour phenotypes predisposed to chemoresistance [21]. Following interaction with DNA, the drug interferes with the gene replication and transcription processes by inhibiting proliferating cells from further division. Indeed, due to the low concentration of chloride in the intracellular environment promoting the aquation process, a chloride ion is gradually displaced by water to give the corresponding aquo-complex. In turn, the water molecule is itself easily displaced by nucleophilic atoms and/or functional groups on DNA to form covalent bonds. Displacement of the residual chloride by another nucleophile can produce crosslinking with consequent irreversible alterations to DNA. Typically, N-heterocyclic bases of DNA are involved in this mechanism. The nitrogen number seven in the guanine nucleotide of DNA is particularly susceptible to alkylation. In addition to the intrastrand crosslink adducts between purine bases (the main type of cisplatin-dependent DNA damage), other adducts include interstrand crosslinks and non-functional adducts that have been postulated to contribute to cisplatin’s activity in DNA damaging [22,23,24]. The formation of DNA-cisplatin adducts elicits repair mechanisms orchestrated by harm recognition proteins, that in turn, depending on the damage type and severity and the repair capacity, can activate apoptosis [19,20]. Over 20 individual proteins could be engaged in this sequence, starting from the binding to physical distortions in the DNA induced by the intrastrand platinum adducts. It is believed that each of the recognition proteins may initiate one or more specific events; thus, DNA damaging results in several unrelated biological effects [25,26,27]. This is consistent with the understanding that adducts induced by cisplatin disrupt replication and transcriptional processes, but that such biological effects do not necessarily correlate directly with cell death [21].

Like other platinum-based compounds, carboplatin and oxaliplatin undergo activation within cells. The resultant reactive platinum complexes exert their cytotoxic effect mostly through DNA damaging, causing intra- and inter-strand crosslinkage of DNA molecules, inhibiting DNA synthesis and transcription, and thereby affecting all the phases of the cell cycle. Apoptosis of cancer cells and immunologic reactions are foregone consequences of these effects [2,5,7,15]. Therefore, carboplatin has also been widely used in the clinic for the treatments of a range of malignancies such as ovarian and lung cancers, but also for cervical, testicular, breast, head and neck, endometrial, and bladder cancers [28]. Although providing the same type of DNA adducts as cisplatin, carboplatin’s structure is more stable, giving the advantages of both lower side effects and longer lasting action compared with cisplatin [2,7,10]. Pertaining to oxaliplatin, it was administered where other Pt-based drugs have not shown therapeutic validity, i.e., in the treatment of colon and colorectal cancers. According to recent reports, oxaliplatin exerts cytotoxicity in cancer cells by causing ribosome biogenesis stress rather than DNA damage [2,7,11,16,29,30,31].

Because of cisplatin and congeners wide use in the clinic, as well as factors deriving from their intrinsic characteristics, increasingly serious drawbacks have gradually emerged over the years that no longer meet the standards of current biomedical research [15,22]. Poor selectivity against cancer cells, severe systemic toxicity, and increasing chemoresistance are just some of the major limitations which are gradually tarnishing the historical success of cisplatin and its derivatives. Indeed, cisplatin action is non-specific and generally targets any kind of dividing cells, leading to extensive side effects, among which include nephrotoxicity and neurotoxicity. In addition, other toxicological effects are very common, such as cardiotoxicity, hepatotoxicity allergic reactions, immunodeficiency, gastrointestinal disorders, and ototoxicity. In turn, the onset of serious side effects forces the therapeutic regimens to be carefully managed in the clinic, thereby inevitably affecting the beneficial efficacy [32]. Carboplatin has, in general, demonstrated lower side effects compared with cisplatin, whereas oxaliplatin did not exhibit excessive nephrotoxicity, ototoxicity, and myelotoxicity; however, a significant dose-dependent neurotoxicity limits its clinical use [10,11,28,29,33,34].

The onset of cancer chemoresistance to platinum-based chemotherapy represents an additional major limitation to its clinical practice. Innate or acquired chemoresistance towards platinum-based anticancer agents is a crucial challenge in modern chemotherapy [21,35]. In-depth studies on the tumour cell phenotypes to be treated should be carried out in advance to obtain the actual benefits of chemotherapy. Accordingly, their clinic efficacy is now limited to some solid tumours, i.e., ovarian, lung, and testicular cancers, whereas many other aggressive and metastatic cancers currently have no therapeutic options [7,15,23]. Chemoresistance to cisplatin and congeners is extensively covered in the next section. Overall, these and other emerging shortcomings are significantly reducing their therapeutic indications and, looking for novel therapeutic options, have gradually provided a major boost to the search for unconventional metal-based drugs.

It is also worth considering that many cisplatin analogues (about thousands of Pt(II) complexes) have been designed and tested to develop safe and clinically effective molecules [14]. However, beyond the best-known ones approved in therapy, such as carboplatin and oxaliplatin, these investigations did not give the desired results. The systemic toxicity persisted as a major challenge and none of the novel analogues have achieved clinical importance in oncology [8,10,14,15,28,36]. The further exploitation of oxidation states of platinum is a modern and promising approach that involves the use of platinum (IV) compounds as next-generation Pt-based anticancer agents, which behave generally as non-toxic prodrugs. Platinum (IV) complexes normally adopt octahedral geometries, producing chemically stable compounds which undergo a slower ligand displacement [2,8,37,38]. As a consequence, these compounds exhibit reduced reactivity towards potential biomolecular targets other than DNA, thereby limiting considerably the onset of adverse effects. The availability of two extra ligands on the metal centre allows for a fine tuning of their physico-chemical and biological features, such as the required degree of lipophilicity, active targeting, immunomodulation, improved cellular uptake, and intracellular trafficking [39,40,41].

Finally, as part of the current research aimed at the progress of platinum-based anticancer agents, in the era of nanomaterials for biomedical applications it is also mandatory to consider the prospective benefits of nanodelivery, which may perchance open new clinical horizons both for old and new generation platinum-based drugs [7,42,43]. Indeed, nanoscale drug delivery devices have already demonstrated significant improvements in the application of some anticancer drugs, such as doxorubicin and vincristine [44]. Correspondingly, lipoplatin—a liposomal formulation containing cisplatin—provided good outcomes in clinical trials and ensures a better tolerability profile than the naked cisplatin. From this perspective, additional ligands of Pt(IV) compounds can be also conceived to allow for the formation of nanosystems [45].

## 2. Chemoresistance to Platinum-Based Metallochemotherapy

The goal of chemotherapy is to selectively eradicate cancer cells by drug-dependent cytotoxicity, which induces their death via activation of specific processes such as apoptosis. Hence, chemotherapeutics should “force” cancer cells towards their self-elimination [20,23,24]. The onset of chemoresistance during clinic regimens represents a critical limitation for the use of chemotherapy. At present, its occurrence represents one of the main hurdles to platinum-based chemotherapy and a primary cause of treatment failure [21]. In addition to significantly reducing drug sensitivity in cancer cells, resistant forces to intensive therapeutic protocols cause inevitable magnification of systemic adverse effects [46]. For this reason, chemoresistance is among the main factors that have fuelled the search for non-platinum metal-based drugs.

At present, mechanisms of resistance have been only partially elucidated. Generally, molecular mechanisms uncovered in preclinical models are consistent with those subsequently detected in the clinic. Considering that biological effects of cisplatin and congeners are triggered by several molecular interactions—from cellular uptake to interaction with nuclear targets and activation of the apoptotic machinery—multiple molecular mechanisms at different levels can partake in resistance development [46]. Thus, resistance onset is considered as a multifactorial dynamic process, with several different pathways active in the same tumour, involving limited drug accumulation, reduced drug-target interactions, increased populations of cancer stem cells, enhanced autophagy activity, and reduced apoptosis [47].

Tumour cells counteract the intracellular bioaccumulation of cytotoxic drugs by interfering with their uptake and metabolism and/or by promoting their efflux through specific membrane transporters. Multidrug resistance proteins (MRPs) metabolize and modify xenobiotic toxic substances, then favour their outflow from the cell [48]. Members of MRPs have been found over-expressed in tumour cells and associated with cellular efflux of a variety of chemotherapeutic drugs. Among these family proteins, MRP2 seems to play an important role in cisplatin resistance [49]. Therefore, MRPs have become a major target of investigations: their expression and activity modulation can re-sensitize cancer cells to chemotherapeutic agents, and MRP inhibitors have been developed recently [50,51,52]. Comparably, a mammalian multidrug resistance (MDR) system mediated by P-glycoproteins (P-gp) acts as an efflux pump system for xenobiotics, including several chemotherapeutics. Indeed, some evidence suggests MDR1 as implicated in multidrug resistance in many tumours [53]. Clinical studies in cisplatin-based treatments have proved P-glycoprotein overexpression associated with a poor chemotherapeutic outcome. Both MRPs and MDR proteins belong to the ABC (ATP-binding cassette) transporter superfamily that are known to translocate various substrates across membranes, including metabolites, lipids, and sterols, as well as xenobiotic drugs [54].

As the intracellular aquation reactions activate cisplatin, several cytoplasmic thiol-containing molecules (e.g., glutathione GSH and metallothioneins) can counteract this process with consequent cisplatin deactivation and resistance development [55]. Conjugation between GSH and cisplatin is in fact accepted as a significant factor in resistance. Experimental data in cisplatin-resistant cancer cell models, verified by clinical evidence, have shown significant increases in cellular levels of GSH. Nevertheless, as in many other cases of resistance, GSH involvement is not necessarily a recurring phenomenon, and this does not allow for the prediction of the development of resistance in the clinic [56].

DNA repair capacity by cancer cells is an additional mechanism of critical importance in determining the onset of chemoresistance. Indeed, platinum complexes exert their cytotoxic effect mostly through DNA damaging, with apoptosis induction being the final effect of DNA adducts production [20,21,22]. A strong correlation between the ability to repair DNA molecular damage and resistance to chemotherapy is nowadays well established, as demonstrated by several reports in human tumour cell lines [57,58]. Numerous cancer molecular mechanisms orchestrated by a number of protein families ensure genomic stability, as well as confer DNA damage tolerance and cell survival. In addition to being associated with cancer progression, deregulation of DNA repair systems may impact the hypersensitivity or resistance to genotoxic agents [59]. Consequently, DNA repair pathways are now believed to be reasonable druggable targets for enhancing tumour sensitivity to cancer therapies [60]. In terms of nuclear and DNA targets, the capability to tolerate platinum-dependent DNA damage is a prerogative of many neoplasms and seems to be linked to p53-dependent regulations, a gene encoding a transcription factor engaged in protection from malignant transformation [61]. Following DNA injury, p53 can induce cell cycle arrest in the G1/S phase to allow DNA repair operations. In the case of repair failure, p53 is one of the main promoters of apoptosis through the recruitment of proapoptotic members of the Bcl-2 family, which in turn act as cell death inducers [62]. Cancer cells endowed with mutations in the oncosuppressive p53 gene exhibit defects in apoptotic functions and can acquire resistance to platinum-based drugs by altering the apoptotic pathway cascade [63]. Furthermore, inhibition of effector caspases in response to chronic treatments with cisplatin and congeners can impair apoptosis. Deregulation of specific signalling pathways in cancer cells hinders the activity of critical caspases (e.g., caspases 3, 8, and 9) engaged in the final steps of the apoptotic process. Indeed, members of the inhibitor of apoptosis (IAP) protein family (e.g., survivin) are highly expressed in most cancers and are frequently associated with a reduced clinical outcome [64].

Key regulations of the apoptotic pathways are orchestrated by the mitochondrial Bcl-2 protein family [65]. Members of this superfamily are engaged in both proapoptotic and antiapoptotic functions (e.g., Bax and Bcl-2, respectively); therefore, they are considered central players in controlling the cell fate decision [66,67]. Indeed, many tumours upregulate antiapoptotic proteins, such as Bcl-2, Bcl-xL, and Mcl-1, and downregulate oncosuppressive ones, such as Bax. Overexpression of Bcl-2 and Bcl-xL is correlated with many cases of cisplatin resistance, frequently in conjunction with increased GSH levels, as well as mutant p53 [68]. Overall, these mechanisms act together to promote the survival of resistant cancer phenotypes. Their in-depth knowledge and interplay will be of outstanding significance for researchers engaged in the search for novel effective strategies to circumvent resistance mechanisms [69]. Cancer cellular rewiring—a complex network of cancer features combined with dysregulated cellular pathways supporting tumour progression—considerably affects the regulation of the apoptotic machinery, leading to evasion from cell death [70]. In this context, Bcl-2 inhibition represents one more challenge towards reprogramming resistant neoplasms to undergo drug-induced apoptosis [66,67,71,72,73]. Moving in this direction, many expectations are placed in non-classical multitarget metal-based agents [74]. As discussed further below, when developed by a rational design approach ad hoc, they could act on different biomolecular targets, revealing a multimodal action capable of both restoring cell death pathways and preventing mechanisms’ underlying chemoresistance [75,76,77].

## 3. Nonplatinum Small Metallochemotherapeutics

Despite the unquestionable therapeutic successes and the world-wide fame achieved by cisplatin and congeners at the at the dawn of the twenty-first century, complications in their clinical use (mainly linked to their poor selectivity and the onset of severe undesirable effects) have increasingly prompted the scientific community towards the search for new anticancer agents [32,34]. Moreover, emerging criticalities, such as chemoresistance as we have extensively discussed, make conventional chemotherapy often ineffective and significantly limit the spectrum of cancers that can be treated with cisplatin and congeners [21,46]. As well as the designing of novel agents among next generation platinum-based complexes, a major route of investigation has become the search for new chemotherapeutics based on transition metals other than platinum [7,8,15,78]. Of course, among the latter, prospective drugs consisting simply of alternative metal centres to platinum have been progressively considered, conceivably by proposing metals sharing many physical and chemical properties with platinum. The use of a specific central metal ion can powerfully impact biological activity since each metal shows distinctive physicochemical features, including redox ability, binding preferences with ligands and targeted molecules, and ligand-exchange kinetics. In this perspective, over the last few decades the platinum-group metals (also known as platinoids, platinum family, or platinum-group elements), clustered together in the periodic table and including ruthenium, rhodium, palladium, osmium, and iridium, have been largely proposed [78,79,80,81,82]. Based on precise molecular knowledge concerning biological targets and action mechanisms, research is making great efforts towards targeted approaches to develop non-classical chemotherapeutic drugs via rational design [83,84,85,86,87]. Overall, and independently of the exact experimental approach, several novel metal-based complexes have been first conceived and then tested for anticancer activity. Unravelling all the literature research products on these topics is not easy. A systematic literature exploration on PubMed (accessed in February 2022) was performed in relation to the last two decades (2001–2021) using specific keywords and subject headings. Among a large body of scientific studies, including both original research articles and reviews, a trend showing a considerable increase over time in the number of science papers per year has been evidenced (Figure 2A). The literature search was then restricted by specific keywords to articles published on “non-platinum”, “non-classical”, or “unconventional” metal-based agents for chemotherapy. Special attention has been devoted to metals such as ruthenium, palladium, and gold, as they are among the most studied ones endowed with potential anticancer activities (Figure 2B,C). Many candidate drugs belonging to this class have been extensively evaluated in preclinical models in vitro and in vivo, but only a few have achieved different stages of clinical studies (Figure 3) [88,89,90]. This proves how biomedical research is active for valid alternatives to platinum-based chemotherapeutics, but also that further efforts will have to be made to gain approval for clinical cancer settings. The shared strategy behind the development of a next-generation of metal-based chemotherapeutics is to overcome the current limits of Pt-based clinical drugs, including toxicity and chemoresistance [91,92,93,94].

By overviewing the complex field of metal-based agents, the recent development of these multi-target anticancer drugs deserves further consideration. Conceived on a rational design, these chemotherapeutics have been designed by a “multitargeted” approach, both to maximize antitumour efficacy and to counteract the onset of resistance [72,86,87]. Since cancer is a multifaceted and multifactorial pathology, the availability of curative options based on single molecules acting simultaneously on multiple targets could represent one of the most promising approaches of forthcoming chemotherapy. To date, several metal-based anticancer agents belonging to this class are in preclinic evaluation and are able to inhibit tumour growth and proliferation through a multimodal model. In turn, this mode of action is based on interactions with various types of molecular targets located in distinct cellular districts [75,95,96]. Generally, these agents are designed to reach different molecular players engaged in cell fate control, such as cytosolic and/or mitochondrial regulatory proteins. One of the main goals is to elicit cytotoxicity by restoring the activity of tumour suppressor genes that induce apoptotic cell death pathways [76,77,97,98]. Moving in this direction, metal-based anticancer drug candidates, such as some ruthenium, palladium, iridium, and gold complexes, are deemed capable of providing new opportunities for the treatment of resistant cancers resulting from multiple pathogenic factors. The variety of metal centres to be explored, their different oxidation states, and the astonishing chemical diversity of the ligands to be selected gives life to molecular platforms endowed with variable geometries and unique electronic, chemical, and steric properties, which can be exploited to accomplish specific biological features, as well as to interact with distinct biomolecular targets. As part of rational approaches to their design, functionalization within specific groups can be optimized to develop candidate metal-based drugs with specific biological activities for precise therapeutic interventions. Moreover, metal complexes can be arranged as neutral, cationic, or anionic based on the nature of the ligands, further enhancing biomolecules targeting via electrostatic or coordinative interactions [6,78,80,84,85,86,91]. Although representing the overwhelming majority of the approved clinical drugs, canonical organic drugs cannot express such a considerable structural and functional repertoire. It is hoped this potential will soon be exploited to test new drugs in the clinic.

In addition to what was already mentioned for platinum-based drugs in the age of nanoscience, the possible transition of oncotherapeutic agents into nanomaterials for biomedical application requires consideration. Though this is not a topic covered herein, nanodelivery could provide further benefits to fully exploit the potential of metal-based anticancer drugs. To date, several reports have showcased that nanoformulations can improve several aspects of metallodrugs under preclinical investigations, i.e., drug efficacy, targeted delivery, and immune activation, as well as biocompatibility and toxicity profiles, thereby surmounting many current drawbacks of classical chemotherapy [90,97,99,100,101,102].

At this point, a special focus is committed to novel anticancer agents based on ruthenium, palladium, rhodium, iridium, and gold complexes, for which more experimental data are nowadays available. The most up-to-date knowledge and progress concerning the inhibition of cancer cell proliferation by a non-platinum metal-based agent, as well as insights into their molecular mechanisms of action in vitro and in vivo through clinical studies, are summarized and discussed.

## 4. Ruthenium-Based Chemotherapeutics

Among the most studied alternative metals to platinum, ruthenium (Ru) is a noble element belonging to the group of platinoids that has been widely employed in the field of medicinal chemistry. An unspecified number of antiproliferative Ru-based complexes has been subjected in the last decades to in-depth preclinical investigations. The literature shows that progress in potential Ru-based anticancer agents is outstanding [99,103,104,105,106,107,108,109] (Figure 2B,C). Nevertheless, despite these efforts only a few Ru-based agents have reached human studies, not without evidence of some concerns, and right now none of these are used in the clinic [103,110]. Therefore, to date, a decisive breakthrough capable of proposing concrete alternatives to platinum-based drugs has not yet occurred. Given the number of Ru-based agents under preclinic development, however, it is expected that further candidates will move into clinical trials. From this perspective, there are many ruthenium complexes with intrinsic features and potential that, beyond question, deserve further development [90,97,98,111,112].

Ruthenium complexes share special properties and the literature on complexes containing this metal centre is exhaustive. It can generate many different types of compounds by ranging in redox states from 0 to +8, and to −2, although the most common are Ru(II), Ru(III) and Ru(IV) derivatives [111]. Ru(II) and Ru(III)—the thermodynamic and kinetic stable forms in physiological conditions—have been largely used in medicinal chemistry for the development of several complexes endowed with interesting anticancer activities, whereas less information is currently available for the less stable Ru(IV) [103,106,107,113,114]. Compared to the square-planar geometry of Pt(II) complexes, Ru(II/III) products are typically hexacoordinated, producing an octahedral configuration, and are able to exhibit different electronic and steric features. The repertoire of eligible ligands allows the formation of a variety of molecular platforms that, biologically speaking, can interact with different targets, providing more modes of action than cisplatin. The structure of the ligands can be exploited to modulate the drug solubility in aqueous biophases, as well as to impact the lipophilicity to control the ability to cross over membranes. In addition, the availability of more labile axial ligands can enable, via exchange reactions, coordination with specific biomolecules to provide for targeted therapeutic effects [103,106,107].

Ru(III) complexes are generally less reactive than Ru(II) complexes and can likely behave as prodrugs undergoing in-cell activation. This property has been exploited to formulate biocompatible and more selective complexes compared to solid tumours [104]. Their reduction to corresponding Ru(II) counterparts implies the in situ formation of more reactive species is accountable for cytotoxic activity. The so-called activation by reduction normally occurs under biological conditions of hypoxia which can be specially found in tumour microenvironments [115,116]. This process can also be affected by cellular reducing agents, such as ascorbate and glutathione, but it remains controversial, and some evidence suggests that anticancer agents based on Ru(III) may be active in this form [117]. On the other hand, some Ru(II) complexes proved to be superior anticancer properties compared to Ru(III) complexes [105,118]. Correlated to possible selectivity, an additional feature making ruthenium(III) complexes promising alternatives to platinum-based drugs is their biocompatibility. Indeed, throughout preclinical tests many Ru(III) complexes have showcased few biological effects towards healthy cells, probably due to their inherent lower reactivity [119,120]. Correspondingly, in animal models the tolerability profiles have proved to be very interesting with respect to cisplatin and congeners [88,95]. Moreover, some researchers postulated the limited toxicity of Ru-based complexes as owed to the chemical similarities between ruthenium and iron, and to the way they interact with many biomolecules, including serum proteins such as transferrin and albumin [121,122]. In particular, the binding to transferrin would reduce the free amount of ruthenium in the plasma and also enhance drug delivery to the tumour site for uptake by cancer cells that frequently overexpress transferrin receptors in response to higher iron demand [123,124]. However, despite these premises it must be considered that throughout clinical trials on Ru-based candidate drugs some limitations relating to adverse effects have still emerged [110].

Among low molecular weight ruthenium coordination complexes, NAMI-A, KP1019, and NKP1339 (Figure 4) are the iconic ones sharing similar structures and having inspired a large platform of ruthenium-based compounds [125]. NAMI-A and NKP1339 have reached clinical trials, but with results to be re-evaluated. Throughout their record, these molecules have been extensively investigated in preclinical trials and considerable information about their mode of action is available both in cellular and animal models. This makes them the most advanced non-platinum metallodrugs, revealing various beneficial properties fitting with rational anticancer drug design and modern biomedical needs [110,121,122,126,127]. The history of these ruthenium complexes began in 1980 when Clarke and co-workers, looking for alternatives to cisplatin structures, performed unprecedented studies on simple hexacoordinated Ru(III) chloroammine complexes [128]. Shortly after, in 1986 the Keppler research group conceived a novel octahedral, water-soluble Ru(III) complex, known as KP418, which showed significant in vivo anticancer activity in colorectal cancer, as well as in tumour models of melanoma and leukaemia [129,130,131]. Over the years, among the various analogues proposed, its indazole derivative named KP1019 has received particular consideration as a bioactive in a broad range of tumours and was subsequently developed into the more soluble sodium salt KP1319/NKP1339//IT-139. The anticancer activity of these complexes in in vivo tumour models was very promising and superior to that of clinical drugs, including cisplatin [132,133,134,135]. Separately, but inspired by these compounds, historically relevant research on ruthenium complexes has been carried out first by Mestroni and Alessio, and then by Sava and collaborators, leading to the development of Ru(III) platforms provided with sulfoxide ligands. Among these, in 1992 the octahedral water-soluble imidazole complex NAMI was selected for anticancer investigations, before being replaced by the more stable imidazolium salt called NAMI-A [136]. At least initially, the literature analysis and development record concerning these Ru(III)-based complexes revealed that a rational design was followed [137]. In animal models NAMI-A exhibited bioactivity that was originally described as antimetastatic because it was focused on secondary tumour lesions and proficient at inhibiting tumour dissemination, coupled with a relatively mild toxicity profile [137,138,139]. Indeed, with respect to cisplatin, first investigations showed NAMI-A as capable at hitting prevalently extracellular targets rather than DNA. Interferences with cell adhesion and migration, as well as angiogenesis, are in line with its presumed antimetastatic activity, detectable exclusively in vivo because the compound is lacking direct cytotoxicity in vitro [120,140,141]. A glance at the literature shows that, in spite of their structural similarity, KP1019 and NAMI-A exhibit rather different preclinical effects. Indeed, KP1019 is active in a broad spectrum of primary tumours, whereas NAMI-A acts exclusively as an antimetastatic agent and is inactive in bioscreen in vitro on different types of human cancer cells [122,125,126,142]. Even what appeared to be an interesting direct cytotoxic activity revealed by some studies in leukemic cell models has been recently downsized [143,144]. Conversely, over the years the cytotoxic activity of KP1019 and of its derivative NKP1319 has been further confirmed in advanced preclinical models. Extensive literature is in fact available on both their chemical features and behaviour in a biological environment [145,146]. Under physiological conditions KP1019 is more stable than NAMI-A, but both undergo hydrolysis of the ligands with formation of poly-oxo species. This process of aqueous biophase and its correlation with the biological activity of these complexes is still a controversial concern [125,126].

NAMI-A was the first Ru-based drug introduced to humans for clinical trials. The study was undertaken starting from 1999 at the National Cancer Institute of Amsterdam (NKI) and reported in 2004 [147]. It was performed on 24 patients with various solid tumours, including colorectal cancer, non-small cell lung cancer (NSCLC), melanoma, ovarian cancer, pancreatic cancer, and mesothelioma. Overall, a stabilization of some patients was noted (especially in the case of NSCLC) but without disease remission, whereas 19 patients (79%) showed disease progression. However, the tolerability profile—the main objective of the phase I study—proved to be much more acceptable than that of other cytotoxic agents, such as cisplatin, with reversible acute events and only mild haematological toxicity. Nevertheless, the occurrence of severe blisters on hands, fingers, and toes was considered as a dose-limiting toxicity [147]. Thus, considering the impressive efficacy of NAMI-A detected in animal models for the treatment of lung metastases, a phase II study was conducted on a cohort of 32 patients with advanced non-small cell lung cancer (NSCLC) by a well-established therapeutic protocol where NAMI-A was used in place of cisplatin in a combined treatment with gemcitabine [141,148]. Unfortunately, both for the emergence of further adverse effects and for the lack of a proven therapeutic efficacy according to the RECIST (response evaluation criteria in solid tumours), the trial was interrupted and declared devoid of positive clues to support further studies.

As much as was discussed, the NKP1339 complex, which is the sodium salt analogue of KP1019 (also known as KP1319 or IT-139), is currently the first-in-class among small Ru(III)-based candidate drugs undergoing clinical trials. It is the most recent representative of this class of compounds and was originally conceived as a precursor in the formulation of KP1019 for clinical testing [146]. Its pharmacokinetic and pharmacodynamic features are widely documented in the literature. As a Ru(III)-based agent, NKP1339 behaves in the bloodstream as a pro-drug, mainly bound to plasma proteins [149]. Animal models were used to gain insight into its distribution and accumulation, showing that significant ruthenium amounts reached several body compartments in an effective way [133]. Compared to NAMI-A, it has in fact higher propensity to cross membranes and to interact with molecular targets after in-cell activation [125,126,134,142]. Overall, much evidence suggests that NKP1339 acts as a multitarget agent that is able to interact with a variety of yet unknown molecular targets, thereby triggering diverse cellular responses, including cell cycle block and DNA damage, but also triggering the induction of oxidative stress and the imbalance of cellular redox homeostasis, as well as mitochondrial damage with consequential pro-apoptotic effects and cell death via the p38 MAPK pathway [111,135,146,150,151,152]. Despite the ability to enter cells, NKP1339-dependent apoptosis can also occur through extrinsic pathway activation, within the frame of a multimodal mode of action that can limit chemoresistance development [153]. Remarkably, suppression of GRP78 (glucose-regulated protein 78) in tumour cells is among the unique mechanisms of action of NKP1339. GRP78 is a chaperone heat-shock protein that recently emerged as the master regulator of the unfolded protein response (UBR) in the endothelium reticulum, typically upregulated in several cancers and associated with intrinsic and drug-induced resistance. Indeed, in vitro and in vivo preclinical studies with NKP1339 have revealed marked synergy in combination with different classes of cancer drugs and overall improved antitumour effects [146,150]. Thus, such a large body of favourable preclinical evidence provides the groundwork for explaining the positive outcome, especially in terms of the safety profile, of clinical trials performed from 2016. The phase I clinical study (Niiki Pharma Inc. (Scottsdale, AZ, USA) and Intezyne Technologies Inc. (Nashville, TN, USA), 2017, NCT01415297) was performed by NKP1339 monotherapy in 46 patients with advanced solid tumours for the evaluation of safety, tolerability, and maximum-tolerated dose (MTD), as well as pharmacokinetic and pharmacodynamic assets. An overall good safety profile with limited evidence of acute systemic toxicity and absence of neurotoxicity heightens the prospects of further in-depth clinical applications for NKP1339 [154]. Upcoming clinical studies could also be planned in combination with other anticancer drugs, since NKP1339 alone has demonstrated in the clinic only moderate antitumour activity [154]. However, like NAMI-A anticancer effects throughout the phase I clinical trial, patients suffering of NSCLC experienced stable disease following administrations, possibly suggesting an enhanced sensitivity of NSCLC to ruthenotherapy. In the context of clinical trials for novel Ru-based drug candidates, preliminary phase I studies with KP1019 must also be included. Indeed, before this small Ru(III)-based complex was replaced by its sodium salt NKP1339, a profitable phase I and pharmacokinetic study was carried out, starting in 2006, by enrolling a limited cohort of patients who experienced disease stabilization with very limited side effects [155,156,157]. Indeed, in contrast to NAMI-A, KP1019 showed a good safety profile, and the maximum tolerated dose was limited only for its solubility. This is the reason why clinical development has been newly redirected to the better soluble NKP1339. Hence, historically KP1019 was the second Ru-based anticancer agent to enter clinical trials after NAMI-A.

Meanwhile, based on these molecular platforms, several other ruthenium-based agents with attractive features were developed as candidate drugs, with the chance of an upcoming “ruthenotherapy”. Moving in this direction, novel NAMI-A derivatives were formulated by the use of different ligands to improve stability and effectiveness. Almost together, in 2012 the research groups of Walsby and Paduano reconsidered a NAMI-A pyridine derivative—named NAMI-Pyr and AziRu, respectively—by replacing the imidazole group with a pyridine ligand, as well as the sodium counterion with imidazolium [158,159]. Due to enhanced lipophilicity, AziRu revealed superior bioactivity compared to NAMI-A on selected panels of human cancer cells, but the overall effectiveness in terms of IC_50_ remained weak to moderate, probably because of low drug intracellular concentration and degradation/instability phenomena in the biological environment. To allow further developments for prospective biomedical applications, the novel AziRu complex was subsequently proposed as a molecular platform for the design of original nucleolipid nanoaggregates endowed with the ability to stabilize and safely deliver this agent to cancer cells. Indeed, by means of their amphiphilic properties, diversely decorated nucleolipidic formulations allow liposome formation in aqueous solutions that ensure protection to the metal core [160,161,162,163]. Subsequently, encouraging results have been achieved in human cellular models of breast cancers, including the aggressive TNBC (triple negative breast cancer) phenotype, where AziRu has provided evidence to act as a multitarget agent by triggering both mitochondrial apoptosis and autophagic pathways [75,104,164]. In this frame, a very recent preclinical study achieved by the authors in a xenograft model of human breast cancer has validated the safety and efficacy in vivo of a unique nanoformulation loaded with AziRu [90,97]. Moving in this direction, in recent years many research teams have developed original nanodevices to improve properties (e.g., stability, solubility, delivery, and cellular uptake) of Ru-based candidate drugs [104,165].

As far as Ru(II)-based complexes are concerned, the RAPTA (ruthenium arene PTA) family, established by the Dyson group in 2005, are experimental drug candidates that are worth a special mention, having demonstrated marked in vitro and in vivo activity against many tumours. They are half-sandwich ruthenium-arene complexes sharing a 3D chemical structure referred to as a piano stool organometallic conformation, containing PTA (1,3,5-triaza-7-phosphaadamantane) ligand(s) [166,167]. Their biological behaviour, as well as anticancer activity, can be tuned by the ligand’s nature around the Ru-arene unit, enabling the design of many derivatives with different degrees of lipophilicity and the potential interaction with a number of biomolecular targets [168]. RAPTAs represent the evolution of Ru-arene organometallic complexes studied since 1972 for their bioactivity [169]. Thereafter, the RAPTA family has become a core of research in the field of non-platinum metal-based agents, and a number of analogues have been conceived and largely evaluated in preclinical studies [91,95,99,106,107,109,127]. Interestingly, despite the differences in the oxidation state of ruthenium, ligands, and final molecular platforms, many RAPTA analogues share many characteristics with NAMI-A, starting from the antimetastatic activity. Among these, RAPTA-C (a *p*-cymene containing derivative) and RAPTA-T (containing a toluene ring) revealed a superior antiproliferative activity in animal models [170]. RAPTA-C induces apoptosis activation both in primary tumours and in metastasis via the p53-JNK pathways, as well as alterations in apoptotic-related proteins such as Bax and Bcl-2 [171]. Moreover, RAPTA-C has demonstrated anti-angiogenic activity in human models of colorectal carcinoma [172]. Tumour angiogenesis is critical for cancer progression and metastasis, and the identification of new molecular targets involved in these mechanisms is a topic of deep investigation. Among the multitude of genomic and non-genomic targets investigated to elucidate mechanisms of action, the ability to interfere with angiogenesis processes is shared by many metallochemotherapeutics, from the Pt-based ones to the ruthenium derivatives. Indeed, we have previously considered antiangiogenic effects underlying the antimetastatic activity of NAMI-A [120,140]. Several endothelial cell functions as well as distinct pathways can be affected, including the nitric oxide synthase (NOS) pathway, which activates the vascular endothelial growth factor (VEGF) and its receptors that are believed to be key factors of angiogenesis [173]. On the other hand, RAPTA-T seems to act mainly as an anti-metastatic agent by inhibiting invasive phenotypes of cancer cells [174]. Several preclinical studies are also currently underway describing the use of RAPTAs in combination with other anticancer drugs, as well as producing advances in the development of the RAPTA family with novel ligands and structures. This research could pave the way for a forthcoming use in clinical trials of the more advanced RAPTA derivatives. Compared to Ru(III)-based complexes that have already approached the clinical stage, RAPTA-C behaves more stably in a biological environment. In this frame, the plethora of preclinical data collected by in vitro and in vivo studies could suggest that RAPTA-C and RAPTA-T are ready to overcome limitations that emerged when the first Ru-based complexes entered clinical trials [127,171].

In the context of the structural diversity of Ru-containing compounds, Gaiddon and co-workers have synthesized and developed several ruthenium-derived compounds (RDCs) containing a Ru(II) atom linked to carbon and nitrogen ligands via strong covalent bonds. Based on the individual design, the electronic behaviour of these anticancer redox organoruthenium derivatives, as well as their reactivity towards biological targets, might be different [127]. RDC11, one of the most bioactive RDCs, proved to be particularly effective in animal models against a wide range of cancer cells with reduced side effects and without being affected by platinum-induced resistance mechanisms. It has been demonstrated that RDC11 alters redox enzyme activity and metabolic pathways, just by inducing limited DNA damages compared with cisplatin. To further explore the mechanism(s) of action of RDC11 and congeners, and to define novel signalling pathways implicated in their anticancer effects, proteomic approaches have been performed, enabling the identification of distinct histones as potential targets able to considerably impact cellular transcriptome and proteome [175].

As with platinum complexes for which new approaches based on different metal oxidation states have been explored, some researchers evaluated the properties of Ru(IV)-based complexes. However, even though the first studies on Ru(IV) complexes date back to 1994, still little information is available on these derivatives as potential anticancer drugs [176]. Their instability due to the high metal oxidation state has significantly limited the use in the medicinal field. However, it is worth noting that there is a novel dual-targeting Ru(IV)-based agent under evaluation in preclinical studies, endowed with antitumour and antimetastatic activity exerted via the PARP pathway and targeting tumour sites through both the enhanced permeability and retention (EPR) effect and transferrin receptor interaction [177]. 

To conclude with Ru-based drugs, a Ru(II) polypyridyl derivative named TLD1433 (Figure 4D) has recently advanced to human clinical trials for evaluation using a photodynamic therapy (PDT) approach in invasive bladder cancer [105,178]. PDT is an expanding area of medicine as a treatment modality for a variety of cancers based on photodynamic effects and dealing with photosensitizer agents including transition metal complexes. TLD1433 is the first Ru-based photosensitizer to have completed successfully the phase Ib trial for bladder cancer therapy (NCT03053635). Meanwhile, approval has been obtained to move TLD1433 to a phase II trial [178]. More broadly, due to their photochemical and photophysical properties, Ru(II) polypyridyl complexes have emerged as suitable photoactive complexes for application in both PDT and, more lately, for photochemotherapy (PCT) [179]. Therefore, on this basis, other Ru(II) photoactive derivatives currently under preclinical evaluation are expected to rapidly advance to the clinic.

## 5. Palladium

Palladium (Pd) is a noble metal belonging to the platinum family and under focus as the central coordinator metal in prospective anticancer complexes. In recent years, the number of Pd-based complexes proposed as an alternative to the classic Pt complexes has increased considerably (Figure 2B,C). Many of them are in advanced preclinical studies, but similar to ruthenium, only a few derivatives have reached the clinic to date [180,181]. The most common Pd oxidation state is +2, although it can exist in other oxidation states. As for other platinoids, the great scientific interest in this metal is due to the similarity in its oxidative +2 state with Pt(II) in terms of both electronic structure and coordination chemistry [92,94]. Indeed, as in patterns already followed in the design of other non-platinum metal-based drugs, Pd-based complexes were conceived starting from the idea of replacing the platinum centre in Pt-based agents to obtain more effective and less toxic compounds. Moving in this direction, potential palladium anticancer drugs were synthesized and tested as early as the 1980s. However, the substitution of Pt by Pd in cisplatin resulted in a compound lacking antitumour activity due to its rapid hydrolysis [92]. More generally, despite platinum and palladium metals sharing many chemical–physical properties, progress in design and synthesis of new Pd(II) complexes has been challenging. This is due to the lower kinetic stability of palladium derivatives compared to the platinum ones. Indeed, Pd-based complexes exchange their ligands much faster than the analogous Pt-based complexes. The high reactivity of Pd-based derivatives matches with both instability in the biological environment and failure to achieve drug targets [180]. Thus, to improve stability in physiological conditions, researchers have envisioned Pd(II)-based complexes in which the central metal ion is bound to strong coordination ligands and/or non-labile moieties [182,183]. Following this path, Pd-based organometallic compounds were found to be particularly stable due to the occurrence of a strong palladium-carbon bond [182,183]. Although the main supposed mechanism of action was DNA damaging, ad hoc designed Pd(II)-based stabilized complexes showed good anticancer activity and reduced toxicity to normal tissues in preclinical studies compared with Pt(II)-based congeners [184,185]. 

Hence, in the last decade a large number of original Pd-based complexes has been investigated in preclinical studies against many types of cancer cells, with the idea of developing new metal-based drugs that limit the side effects of current treatments with cisplatin or related compounds. Given the number of structures that can be found in the literature, a systematic classification of Pd-organometallic derivatives is rather difficult. Scattolin and co-workers have recently reviewed this topic and attempted to classify Pd-based complexes based on their main structural characteristics. Pd(II)-based chemotherapeutics have been hitherto classified into mono- and polynuclear Pd(II) complexes. The first ones are conceived with one Pd(II) atom in their core, whereas the second ones include more palladium atoms—generally two—in their structure [180]. A palladium core can be bound to different types of ligands accounting for activity and toxicity. Drug design for square-planar mononuclear palladium complexes includes active ligands to confer biological effects, water-soluble ligands, and strong coordination ligands. Generally, the primary target of Pd(II) complexes seems to be DNA. It has been reported that they are able to bind to both covalently and non-covalently DNA in tumour cells, causing genetic damage and inhibition of duplication [182]. Since Pd-based complexes are recurrently active in cisplatin-resistant cells, it has been also assumed that the interaction with genetic material can occur in different ways with respect to cisplatin, mainly involving non-covalent interactions [186]. Molecular characteristics of the Pd(II)-based agent significantly influence drug–DNA interactions, with aromatic planar moieties as the most common ligands to promote DNA damage [182]. However, a number of different classes of ligands was used to synthesize Pd-based complexes to be tested in vitro for preliminary evaluations [187]. Another proposed target for Pd(II)-based complexes is mitochondria, resulting in the activation of both intrinsic and extrinsic apoptosis cell death pathways. The onset of intrinsic apoptosis is often associated with an inversion of a Bax/Bcl-2 ratio, the release of cytochrome *c*, and subsequently, caspase cascade activation, whereas extrinsic apoptosis activation is triggered by an increase in the expression of cell death receptor genes DR4 and DR5 [183,185,187,188]. It has been also demonstrated that some Pd(II)-based complexes can block the cell cycle at the G2/M phase, and other Pd(II) complexes can interact with intracellular sulfhydryl proteins’ groups, such as those of the antioxidant systems. These interactions are associated with an increase of intracellular ROS followed by redox imbalance and oxidative stress, as demonstrated by reticulum endoplasmic alteration in cancer cells treated with these compounds [189]. Some derivatives of Pd(II) have shown interesting anticancer effects in specific cellular models, such as leukemic ones [190]. Moreover, an activity higher than that of platinum derivatives was emphasized in preclinical models of BC, including anticancer activity and putative pharmacological targets towards TNBC [191]. The latter certainly represents one of the most interesting fields for expansion in Pd(II)-based complexes, given the spreading and heterogeneity of this tumour pathology. 

In parallel, several mono- and dinuclear organometallic Pd-based complexes endowed with non-canonical original structures and known as palladacycles have been conceived and reported for their preclinical activities, showing potential for therapeutic use. They can be synthesized using bidentate bridging ligands (e.g., bidentate phosphine ligands), together with additional ligands selected for their capability to trigger interactions with a biomolecular target [192,193]. Several of these derivatives showed higher anticancer activity than the mononuclear complexes when tested in vitro and in vivo against different human tumours associated with several types of mechanisms of action, including DNA damage, inhibition of cancer cell metabolism, mitochondrial dysfunctions, apoptosis, and autophagy, as well as angiogenesis modulation [185,194,195]. More recently, a novel binuclear palladacycle derivative referred to as AJ-5 has shown in various experimental models in vitro to be particularly active, with a good safety profile in vivo. As well, this candidate metallodrug causes DNA damage with the activation of multiple cell death pathways. A very promising activity deserving further investigation has been reported in cellular models of oestrogen-receptor-positive (MCF7) BC, in TNBC (MDA-MB-231), and in BC stem cells [196]. AJ-5 showed relevant activities also in preclinical models of melanoma [197]. Furthermore, an interesting newer study that supports AJ-5 has a favourable therapeutic agent for the treatment of different sarcoma subtypes, which are among the most common soft tissue cancers mainly spread in the young population and with few remedial options. Following treatment with this organopalladium compound, rhabdomyosarcoma cells undergo cell cycle arrest, reduction in autophagic flux, and induction of apoptosis, confirming a promising pharmacokinetic and toxicological profile in animal models [198]. On this ground, several other dinuclear palladium complexes have been proposed, some of which are undergoing preclinical evaluation [180,181,193,199]. 

Organometallic palladium–saccharinate complexes represent another class of Pd(II) derivatives that has been studied [200]. They exhibit considerable antiproliferative activity against different cancers in both in vitro and in vivo models. In particular, compelling antitumour activity has been described against BC through cell death activation via apoptosis [201]. In oestrogen-responsive MCF-7 and triple-negative MDA-MB-231 cell models, experimental IC_50_ values were in the low micromolar range. Proteomic analysis revealed the modulation of many proteins involved in different metabolic pathways including apoptosis [202]. Promising antiproliferative activities via apoptosis induction in preclinical BC cellular models, including cisplatin-resistant cells, have also emerged during the trialling of binuclear Pd(II) complexes, conceived using biogenic polyamines (e.g., spermine) and named Pd2Spm. In addition to showing additional antiangiogenic and antimetastatic activities in vivo, Pd2Spm derivatives seem to share profitable pharmacokinetic and toxicological profiles; thus, additional investigations towards prospective clinical applications are warranted [203,204]. These findings still confirm attractive bioactivities for many Pd(II) derivatives working against BC tumour models, including TNBC phenotypes [191].

As for ruthenium, a decisive breakthrough has not yet occurred for Pd-based complexes [92]. Comparably, despite the quantity of compounds which have been synthetized and developed for preclinical experimentation, only one Pd-based drug is currently in the clinic; moreover, it is similar for photodynamic applications. Padoporfin (palladium bacteriopheophorbide; WST-09; Tookad^®^), together with its soluble variant padeliporfin (Figure 5) (palladium bacteriopheophorbide monolysine taurine; WST-11; Tookad^®^Soluble) were developed from bacteriochlorophylls and contain palladium as the central coordinator ion [205]. They are used as a sensitizing agent for focal vascular-targeted photodynamic therapy (PDT) (or vascular targeted photochemotherapy, VTP), a minimally invasive treatment procedure requiring local activation by exposition to low-power laser light after administration [206]. The development of prostate cancer PDT has enhanced quickly so that, following clinical trials for the therapy of localized prostate cancer, padeliporfin was commercialized in 2017 with the name of TOOKAD as the first Pd(II)-based compound used in the clinic, demonstrating hitherto to be a safe and well-tolerated photodynamic agent [207,208]. In addition, padeliporforin is currently being studied for the treatment of some kidney neoplasms [209].

Retracing the attractive road of Pt(IV) complexes as potential anticancer drugs, Pd(IV) complexes have been synthesized and tested as well. Although their activity has been reported to be promising, difficult synthetic procedures and a chemical stability that is yet to be verified have thus far limited their further development [210]. Alternatively, low-valent Pd-based complexes have progressively attracted attention within drug discovery programs. It has been recently reported that organometallic Pd(0) derivatives containing purine-based N-heterocyclic carbenes exert significant antiproliferative activity against ovarian cancer cell lines, comparable to cisplatin even in resistant cell lines [211]. Moreover, following an evolution of Pd-based derivatives from the more common Pd(II) complexes to the Pd(0) ones, Scattolin and co-workers reported in 2020 about high and selective antiproliferative activity on different cancer cell lines and advanced preclinical ovarian cancer models of an original dinuclear Pd(I)-based complex, showing mechanisms of action different from that of cisplatin and mainly involving mitochondria [212]. These findings could pave the way towards the discovery of new alternatives among organopalladium derivatives endowed with antiproliferative action. 

## 6. Rhodium and Iridium

Belonging to the group of so-called platinoids, rhodium and iridium have been recently proposed as central coordinator ions of several metal complexes conceived and designed for antitumour therapy. Among them, some experimental drugs have been reported as selective and effective against different cancer cells, showing original mechanisms of action compared to classical chemotherapeutics [213]. Nevertheless, from this standpoint, they have a rather limited story. However, in a historical general overview, it should be noted that the anticancer properties of rhodium were explored before the discovery of cisplatin, as reported in a study way back in 1953 concerning a simple Rh(III) complex [214]. At the end of the last century, research on antitumour complexes aroused some interest around both dimeric and square-planar Rh(II) and Rh(I), endowed with a geometry similar to that of cisplatin and showing bioactivity based on the ligand-exchange process. Six-coordinated cage di-rhodium(II) tetracarboxylate complexes and their derivatives were studied with interest but not further developed due to the appearance of toxic effects [215,216]. Investigated antitumour Rh(I) compounds were the organometallic neutral and square-planar cyclooctadiene complexes. Both were found bioactive in vitro and in vivo against many tumours, such as leukaemia, oral carcinoma, melanoma, mammary carcinoma, Lewis lung carcinoma, and lung metastatic tumours. Their mechanisms of action have not been explored systematically, but an interference with proteins regulating the metabolism of nucleic acids seems feasible [215,216]. Although they have been studied for a long time, few complexes of this type are now being tested and are much less discussed. Conversely, the literature of the last decade shows rhodium and iridium in their +3-oxidation state as a growing concern in biomedical research, but still very far from the progress levels reached by ruthenium-based agents [89,217]. Thus, considerable efforts will have to be made to move Rh- and Ir-based experimental drugs forward in clinical studies. In their favour, higher biological activity, water solubility, stability, and simple synthetic procedures are factors that could provide for further advances [213]. As well, with respect to the square-planar Pt(II) complexes, the molecular octahedral architecture of both Rh(III) and Ir(III) complexes confers structural multiplicity and unique properties in feasible biomolecular targeting. Suitably functionalized, these complexes can in fact act as inhibitors of proteins or modulators of protein–protein interactions. In contrast, rhodium and iridium metal centres share chemical inertia and slow kinetics in a ligand exchange, which make their use more problematic for biological targeting. In comparison, ruthenium(II/III) complexes’ ligand exchange rate is roughly in the same range of that of platinum(II) ligands (10^−2^ to 10^−3^ s^−1^), allowing for the design of numerous reactive agents [218]. However, the insertion of proper ligands and structural modulations in organometal Rh(III) and Ir(III) complexes can significantly enhance their reactivity [213,217]. 

With exclusive reference to the antitumour activity with the most recent progress, functionalized Rh(III)-based complexes designed to reach specific targets are in preclinical evaluations with very encouraging results. Topoisomerase II α (topoII α) is a key enzyme for DNA replication representing a potential druggable target for numerous types of neoplasms to produce therapeutic effects. Rhodium complexes functionalized with thiomaltol moiety have shown interesting antiproliferative activities and induction of apoptosis. Some of these derivatives also caused impairment of cellular redox homeostasis correlated with ROS generation [219]. Following this path, similar organometallic derivatives were prepared with other platinoids, i.e., ruthenium, osmium, and iridium [220]. An additional, latest line of research engaged in the development of new, promising Rh(III)-based complexes with antineoplastic action is embodied by protein kinase inhibitors. Phosphorylation is a crucial regulatory process in a number of metabolic and signal transduction processes. As a consequence, kinase inhibitors can act as key modulators to manage a variety of diseases including different types of cancer. From this perspective, Rh(III) has proven to be suitable as a metal centre for the design of octahedral organometallic platforms, which are promising scaffold candidates for protein kinase inhibition [217]. Among potential antitumour agents under investigation, some novel cyclometallated Rh(III) complexes have proven to be good inhibitors of Wee1, a tyrosine kinase belonging to the Ser/Thr protein kinases family regulating proliferation via mitosis timing, as well as checkpoints involved in cell growth and proliferation. Wee1 inhibition by Rh(III)-based complexes has shown potential in limiting growth in cellular models of triple negative breast cancer [221]. Remaining in this framework, other inert but ad hoc functionalized Rh(III)-based complexes containing bipyridine ancillary ligands have shown inhibitory activity against mTOR kinase, which is engaged in several metabolic regulations controlling cell division and differentiation, but is also an important druggable deregulated target in many human tumours [217,222]. The same activity has been documented for some Ir(III) complexes functionalized in the same way [217,222]. Thus, organometallic platforms developed from a central ion belonging to the platinoid family represent the first metal-based class of prospective inhibitors of mTOR activity and proves Rh(III)- and Ir(III)-based agents as potential protein modulators. Moving in this direction, other Rh(III)-based organometallic protein inhibitors have been recently developed including molecules interfering with thioredoxin reductase (TrxR) activity. TrxR is frequently overexpressed in human tumours as a regulator of intracellular redox homeostasis that enables evasion of apoptosis in cancer cells and represents another possible druggable protein target implicated in the development and progression of cancer [223]. As well, high cellular levels of TrxR, together with the substrate Trx, have been associated with resistance to cisplatin [224].

Among non-platinum-based scaffolds exploited as prospective anticancer candidates, Ir-based organometallic derivatives represent an emerging class of drug candidates. Although still discussed less than rhodium derivatives, the literature of the last decade shows considerable and expanding interest in this noble metal. Indeed, research around its anticancer complexes has great potentiality, the most attractive being the newest highly versatile Ir(III)-based ones endowed with half-sandwich octahedral geometry. As for Rh-based complexes, the first organoiridium complexes to be explored were square-planar Ir(I) complexes because of their structural and electronic similarity to Pt(II) anticancer complexes, such as cisplatin and its derivatives [225,226]. Several Ir(I) N-heterocyclic carbene (NHC) complexes have attracted interest, but many of their biological properties are still unexplored. To provide for a first structure-activity relationship, it has recently been reported that distinct Ir(I)-based complexes can exert cytotoxic effects via selective interactions with biologically relevant proteins, such as cytochrome *c*. Interestingly, this molecular interaction is coupled with the oxidation of the metal centre from Ir(I) to Ir(III), as part of a process that the authors have defined as “oxidative protein binding” [227]. Then, in recent years the focus moved on to the more easily tuneable organometallic Ir(III)-based products which, at least in theory, could permit a consistent number of biomolecular interactions by appropriate functionalization with ancillary ligands [225]. Moreover, their ligand substitution kinetics are orders of magnitude larger than those of platinum complexes; therefore, they are particularly suitable as inert scaffolds for the design of specific protein inhibitors [217]. In this frame, it is likely that some Rh(III) organometallic, ad hoc conceived Ir(III)-based candidate drugs can target mammalian TrxR and exert their cytotoxicity in tumour cells [228]. However, nowadays evidence suggests many Ir(III)-based complexes are acting as anticancer agents via the targeting of apoptotic pathways [77,229]. By specific incorporation of binding ligands, the latest preclinical advances revealed that organoiridium(III) complexes exhibit higher cytotoxicity versus several human cancer cells (e.g., breast, colon, prostate, melanoma, and leukaemia) compared to cisplatin, probably by DNA interactions and redox homeostasis perturbations [230]. Moreover, in the last years, different research groups have synthesized several Ir(III)-based complexes with various functional moieties, such as half-sandwich or cyclometallated pseudo-octahedral derivatives, exerting remarkable mitochondria-targeted anticancer activity. Rather than causing DNA binding and damaging, these derivatives seem to upregulate and restore apoptosis pathways in cancer cells via cellular redox imbalance and ROS production, as well as via mitochondrial dysfunction and membrane potential fluctuations [231,232]. Comparably to the Ru(III)-based AziRu complex, some N-heterocycle derivatives of Ir(III) metallodrugs have showcased superior in vivo anticancer effect that is also associated to autophagy-regulating activities [233]. The regulation of autophagic pathways in cancer cells can in fact open new opportunities in the design of chemotherapeutic strategies to block tumour proliferation [97,164].

## 7. Gold

Au-containing compounds have been widely employed in various fields of medicine throughout history. Though not belonging to the platinum family, Au-based derivatives deserve consideration in the universe of metal-based chemotherapeutics [234,235]. Indeed, although clinical applications have thus far concerned the treatment of rheumatoid arthritis (RA), they represent an emerging class of non-canonical anticancer metallodrugs, endowed with effective biological activity [78,92,93,236]. As shown in Figure 2B, in the last decades the total amount of research products on topics concerning the development of Au-based anticancer agents are numerically second only to the Ru-based ones. Furthermore, the search hits timeline illustrated in Figure 2C reveals that in recent years the total number of “cancer papers” focusing on gold derivatives are the most published ones. 

Gold can exist in several oxidation states (from −1 to +5), with Au(I) and Au(III) derivatives as the most investigated prospective anticancer compounds. The first are “soft” metal centres that have strong tendencies to form stable complexes with easily polarisable ligands, such as sulphur or phosphorus-containing groups; the latter are “hard” metal centres with a preference to form more reactive tetra-coordinate square-planar geometry complexes with oxygen- or nitrogen-based ligands [237]. Of course, the search for Au-based compounds as promising anticancer agents started from studies on Au(I)-containing molecules already known for their therapeutic properties and used as such (e.g., auranofin and aurothiomalate) or as inspiration for the design of safer molecular structures. Developed from about 1920, gold-based therapy referred to as aurotherapy or chrysotherapy has been for a long time the main treatment for RA or other inflammatory conditions aiming at reducing inflammation and disease progression. Auranofin was approved by the FDA in 1985 as a therapeutic agent to target rheumatoid arthritis as opposed to parenteral use of conventional treatments. Indeed, auranofin is of special interest since it can be administered orally in contrast to the other Au-based drugs [237]. Aurotherapy application progressively decayed due to various factors including the onset of toxic side effects (e.g., liver toxicity, kidney damage, and bone marrow diseases). This research trend, as part of a drug repurposing program, has allowed auro-derivatives such as auranofin and aurothiomalate (Figure 6) to enter clinical trials for the treatment of some human cancers. Original experimental research by in vitro and in vivo models unveiled auranofin as displaying inhibition of cancer cell growth [238]. Although preclinical applications revealed constant efficacy in vivo only against leukemic cells, they were the launching pad for Au-based compounds in cancer chemotherapy. The reported biological effects are likely the result of TrxR inhibition, which occurs with high potency and selectivity on both the cytosolic and mitochondrial forms of this selenoprotein [239]. TrxR druggability as a biological target in human cancer has been already discussed in relation to metal-based chemotherapeutics other than gold [223]. Selenocysteine residues in catalytic sites of TrxR are critical for the enzymatic activity, such that a covalent interaction based on the electrophilic Au(I) centre and the nucleophilic sulphur residues seems to underlie the drug molecular mechanism of action [239]. In Jurkat cells, inhibition of both cytosolic and mitochondrial TrxR is associated with oxidative stress by enhanced hydrogen peroxide levels, ultimately resulting in apoptosis induction [240]. Consequently, findings from these studies suggested a general mode of action for gold agents via inhibition of cysteine or selenocysteine containing enzymes [241]. Thus, starting from auranofin, an expanding number of investigations on the potential of Au(I)-based complexes as anticancer drugs have focused on analogues of this compound. There are probably hundreds of analogues synthesized and tested in preclinical screenings. Some of them showed activities higher than those of cisplatin in in vitro models [237,242]. Gold(I)-phosphine derivatives (including complexes with multiple phosphine ligands) are the most numerous and studied, but compounds designed by means of different ligands showing interesting activities cannot be underestimated [243]. For instance, gold–phosphole compounds conceived by a phosphacyclopentadiene ligand attached to the central metal have demonstrated the ability to inhibit cancer growth in vitro that is associated to strong inhibition of TrxR and the related glutathione reductase [241,244]. Given the repertoire of structures and ligands that Au(I)-based species under evaluation share, the assumption of a single mechanism of action would be unreasonable. Of course, many biological features of these compounds rely on the type of ligands attached to the metal centre, directly impacting their targeting [245]. For some of them even an interaction with DNA cannot be excluded [236,237]. Since these compounds derive from drugs applied for anti-inflammatory therapies, actions on other classes of enzymes such as cyclooxygenases (COX) and lipoxygenases (LOX) have been also evaluated [246].

Au(III)-based complexes derive from more recent investigations encouraged by their structural analogies with cisplatin and congeners, and nowadays, they are probably the most promising emerging class of new Au-based anticancer agents. From this perspective, considerable efforts are underway looking for stable and effective derivatives (by ligand design) in the biological environment. In fact, one of the main challenges encountered during the development of Au(III)-based complexes was their instability in physiological conditions via intracellular redox reactions [234,235,236,237]. The design and development of chelating ligands such as nitrogen donors, cyclometallated structures, and dithiocarbamates has allowed for the production of definite metal-based complexes suitable for biomedical applications. For these complexes, significant bioactivity both in in vitro and in vivo models has been observed and only recently are indications concerning mechanisms underlying their antiproliferative activity beginning to emerge [247,248]. The recent literature highlights an additional strategy to develop more bioactive Au(I) and Au(III) complexes, based on the selection of specialized ligands holding biological/antitumour activity. Hence, as already occurred for other metal-based drugs, a future and promising direction for novel auro-derivatives’ design encompasses the evaluation of ad hoc functionalized multitarget agents able to hit tumour cells by multiple mechanisms of action [74]. An outgrowth of these efforts has been the design and synthesis of a wide range of Au-containing species with encouraging anticancer properties, comprising Au(I)-based phosphane derivatives and thiolates, gold(III) complexes with porphyrin-type or bipyridyl-type ligands, and gold(III) organometallic and cyclometallated compounds [249,250,251,252,253]. Their potential is of special interest as they have demonstrated higher selectively and efficacy with respect to Pt-based complexes. Efficacy has been verified by preclinical bioscreens on selected panels of human cancer cells, and some derivatives have proved to be excellent drug candidates for future clinical applications [236,237]. Based on the great structural variety of the used ligands, a unique mode of action or pharmacological profile is unlikely to exist. In general, Au(III) complexes have a greater tendency than Au(I) species to interact with DNA, but in turn their ability to interact with DNA remains much lower than that of cisplatin and its congeners. Thus, within a context of non-cisplatin-like pharmacodynamics, Au(III)-based chemotherapeutics have proved to inhibit cancer cell proliferation through a variety of DNA-independent mechanisms [247]. Like other auro-derivatives, they revealed the ability to interact with thiol groups which in turn empowers association with a number of cellular enzymatic components, including the selenoenzyme TrxR. In fact, the most relevant biological effect proposed for Au(III) derivatives is still the inhibition of TrxR. Alterative mechanisms have been also proposed over the years to support cellular-specific antiproliferative effects, i.e., the inhibition of topoisomerase, targeting of mitochondrial functions, apoptosis induction, proteasome inhibition, and modulation of specific kinases [78,254,255]. It is noteworthy that recent studies focusing on innate and adaptive antitumour immunity have explored the effects of a possible reversal in cancer cell immune escape by both Au(I) (including auranofin) and Au(III)-based agents via a direct action on immune cell functions, leading to an overall improved anticancer activity. In this new scenario, the suppression of cancer-promoting inflammation has been assumed as one of the main mechanisms exerted by Au-containing compounds. Pondering their synthetic flexibility, the potential combination of inherent antitumour activity with prospective immunomodulatory effects throughout the Au-based metal complexes’ design could represent a new frontier in the development of effective metallochemotherapeutics for cancer therapy [256,257].

Nevertheless, except for the old antirheumatic drugs recently reproposed as anticancer drugs, and considering the number of unique Au-based compounds conceived and tested thus far, no one has yet reached clinical trials. The most demanding challenge in this field is the design of species with a stable behaviour in the biological environment, as well as with appropriate pharmacokinetic characteristics to reach cellular targets. These features depend on a multitude of factors mainly related to the oxidation state of the metal centre and to the variety of ligands needed to originate the final molecular platforms. Moreover, the elucidation of cellular responses associated with detrimental effects also represent an urgent concern to allow for the further development of aurodrug candidates towards the future transition to the clinic [236,248].

As far as clinical trials are concerned, Auranofin (Figure 6A) has proved potential to be repurposed for malignant disease [258]. It has proven anticancer activity in animal models and, although many aspects remain to be clarified about its mechanism(s) of action, Auranofin hinders the TrxR system that is critical to avoid cellular redox imbalance. Its deactivation in cancer triggers oxidative stress, followed by apoptotic death. It is no coincidence that TrxR overexpression is associated with aggressive tumour progression and poor survival in patients with breast, ovarian, and lung cancers [259,260]. Moreover, Auranofin was recently reported to interfere with proteasome activity, another attractive druggable target under evaluation for new anticancer strategies [261]. To date, Auranofin has entered different clinical trials as an anticancer drug for the treatment of ovarian, glioblastoma, and chronic lymphocytic leukaemia (NCT01747798; NCT02126527; NCT03456700; NCT01737502; NCT02063698; NCT01419691; NCT02770378). In addition, phase I/II trials are exploring the safety and toxicological profile of auranofin when presented together with sirolimus (a macrocyclic antibiotic with potent immunosuppressive activity) in treating patients with advanced or recurrent non-small cell lung cancer or small cell lung cancer (NSCLC and SCLC) without standard treatment options (NCT01737502). Consistently, following positive outcomes in preclinical models suggesting potent anticancer activity, in 2007 aurothiomalate (ATM, Figure 6B) has advanced to a phase I trial in patients with NSCLC, ovarian, or pancreatic cancers (NCT00575393). This clinical study was completed in 2018. ATM has demonstrated to inhibit biosignalling mediated by the protein kinase Ciota (PKCι), which is upregulated in many human cancers [262]. 

## 8. Concluding Remarks and Future Perspectives

According to the data available from the WHO, cancer is a leading cause of death worldwide, accounting for nearly 10 million deaths in 2020. Such a human burden embodies a great stimulus for modern researchers engaged in the uncovering of safe and effective remedies for the treatment of this complex pathology. From the viewpoint of pharmaceutics, great progress has been made, but many goals remain to be achieved, especially to defeat the metastatic disease where chemotherapy is the only functional weapon. Though almost all the drugs currently in use derive from carbon chemistry, metal-based complexes have always been of special interest in cancer therapy. Indeed, since their approval in the clinic, cisplatin and congeners have represented lead drugs and are still reference drugs in the treatment of some human cancers. Following this path, the concern in metallodrugs has expanded considerably over time, to the point that today numerous transition metals other than platinum are used in the design of new potential anticancer drugs and more. As evidenced by the scientific literature, one of the most fashionable options is to design platinoids to develop innovative anticancer agents endowed with molecular mechanisms of action and clinical profiles different from Pt-based drugs. The herein reviewed shared strategy underlying the advancement of the next generations of metal-based chemotherapeutics is to overcome the current limits of Pt-based clinical drugs, including toxicity and chemoresistance. Depending on a huge variety of factors throughout their design, non-classical metal-based compounds can give rise to molecular platforms and chemical spaces potentially skilled at interacting with an indefinable number of molecular targets. In this frame, the multi-targeted approach represents one of the most promising in order to increase selectivity and efficacy towards specific cancer phenotypes. Indeed, the concurrent activation of multiple cell death pathways could significantly decrease the development of chemoresistance, which plagues many current therapies. From this point of view, metal-based drugs offer unique opportunities thanks to their synthetic versatility and ligand selection. In the coming years this potential must be exploited both academically and industrially, primarily to select from the massive quantity of platinum-free derivatives in preclinical investigations the most promising metal-based drug candidates to advance to clinical stages. The very small number of compounds having reached clinical trials, compared to the amount of ruthenium, palladium, and gold derivatives under investigation, just to name a few examples, represents nowadays one of the main drawbacks in the field of non-platinum metallodrugs. In our opinion, advanced preclinical models rightly homologous to human cancers will play an increasingly critical role in this process. Collection of relevant and reliable preclinical data can in fact considerably improve the selection of new, effective, and safe drug candidates and make their bioscreening faster and more accurate towards the transition to the clinic. Upstream from the entire process, mechanistically driven drug discovery based on biochemical and pharmacological deep knowledge can considerably impact the development of metal-based agents. High-throughput screening (HTS) is now a well-established process for lead discovery through large-scale data analysis in pharma and biotech companies, but also increasingly applied in academic research. As well, multiple computational approaches for rational drug discovery can significantly restrict the number of active compounds to be screened in preclinical studies. Thus, looking for targeted bioactivity, innovative and technological approaches will be progressively critical for accurate identification of potential candidate drugs, making new and effective chemical weapons available for cancer patients. 

## Figures and Tables

**Figure 1 pharmaceutics-14-00954-f001:**
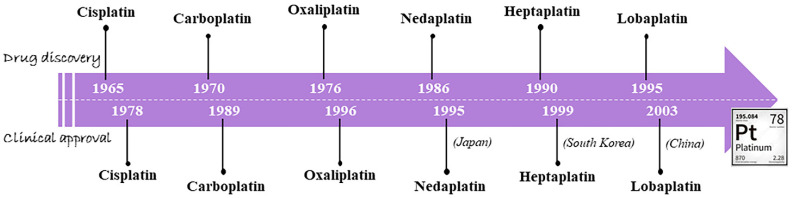
Timeline depicting the discovery of the main Pt(II)-based anticancer drugs and their subsequent approval for clinical use.

**Figure 2 pharmaceutics-14-00954-f002:**
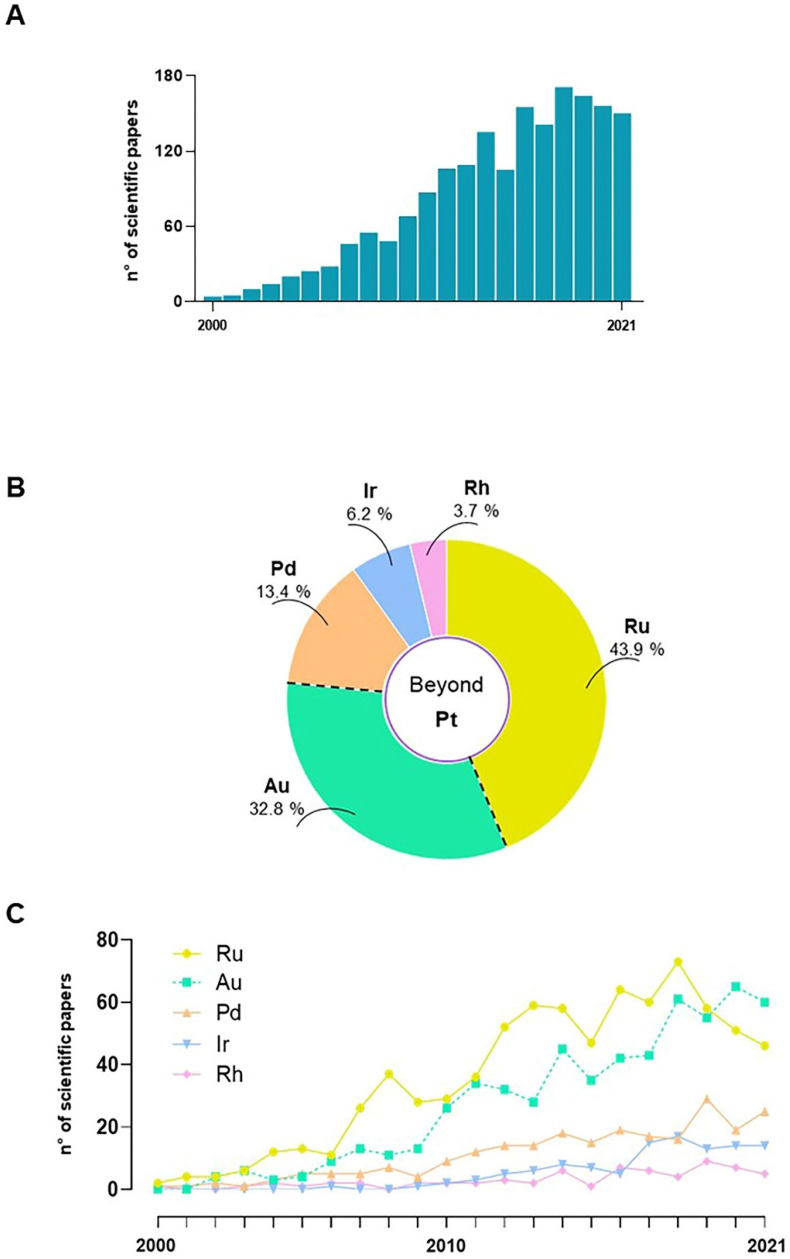
(**A**) Results by year over the past two decades (2000–2021) of scientific studies pertaining to metal-based anticancer drugs obtained through the PubMed database (accessed in February 2022). The analysis was performed by using specific keywords and subject heading strings combined using the Boolean operator “AND” (e.g., “metal-based anticancer drug” AND “metal-based anticancer agents” AND “metal-based anticancer compounds”). (**B**) The pie chart shows the percentage of science papers with respect to the total related to Ru-based, Au-based, Pd-based, Ir-based, and Rh-based anticancer agents in the PubMed database (accessed in February 2022) in the period from 2000 to 2021. The query was formulated with the single metal name combined with the words “anticancer” and “antitumoral” by the use of the Boolean operator “AND”. (**C**) The graph illustrates the search hits timeline from 2000 to 2021 correlated to scientific papers focused on Ru, Au, Pd, Ir, and Rh as metal-centres of anticancer agents traceable in the PubMed database. The analysis has been achieved by combining the names of the respective metals with the words “anticancer” and “antitumoral” using the Boolean operator “AND”.

**Figure 3 pharmaceutics-14-00954-f003:**
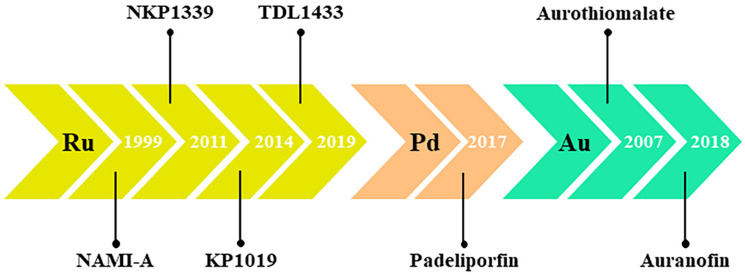
Non-platinum metal-based drug having entered the stage of clinical trials.

**Figure 4 pharmaceutics-14-00954-f004:**
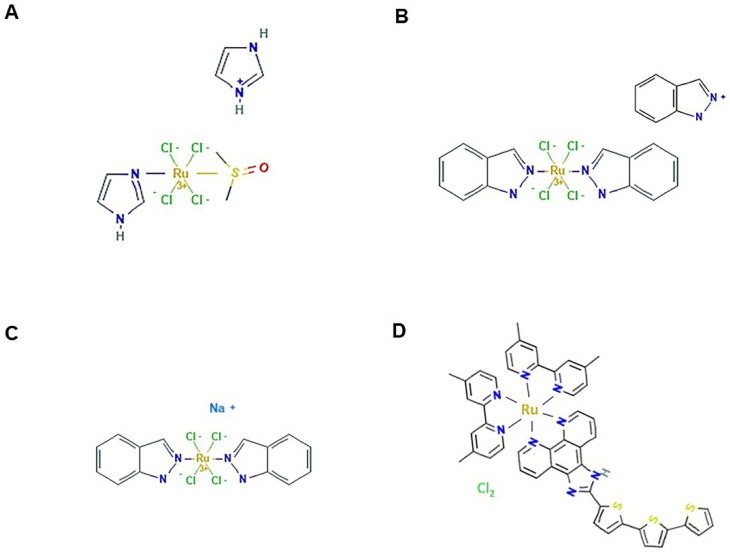
Ru-based compounds evaluated in clinical trials. (**A**) NAMI-A, (**B**) KP1019, (**C**) KP1319/NKP1339/IT-139, and (**D**) TLD 1433.

**Figure 5 pharmaceutics-14-00954-f005:**
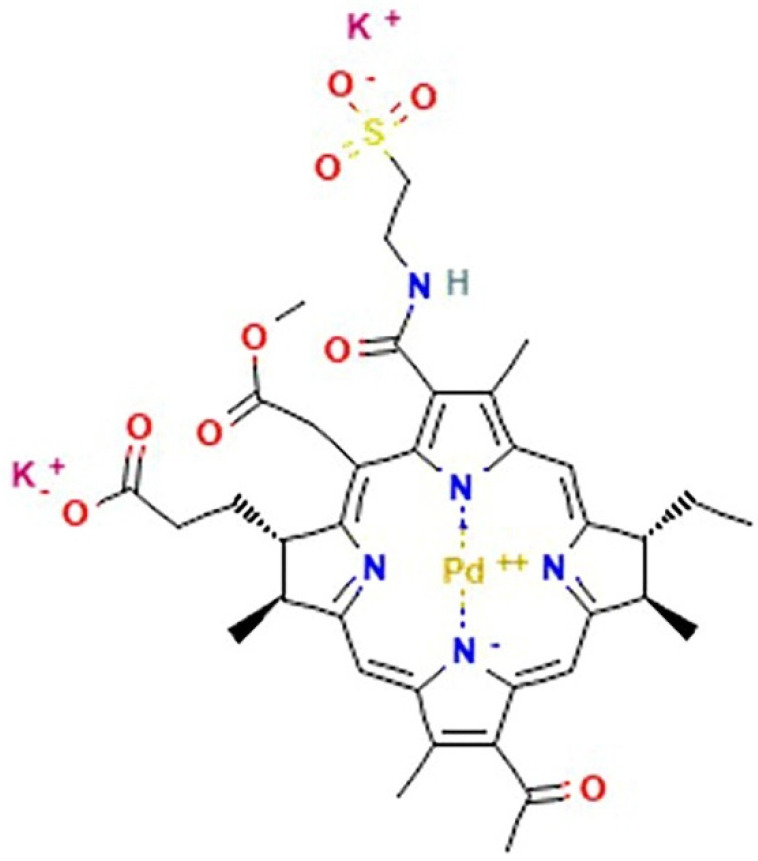
Padeliporfin, the first Pd(II)-based compound used in clinic as photodynamic agent.

**Figure 6 pharmaceutics-14-00954-f006:**
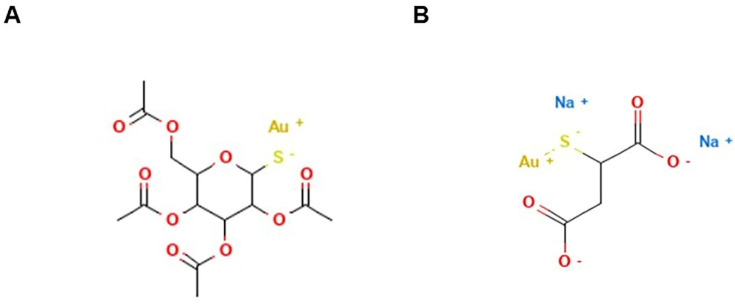
Au-based drugs currently in clinical trials. (**A**) Auranofin, (**B**) Aurothiomalate.

## Data Availability

Chemical Structure depictions in Figure 4, Figure 5 and Figure 6 are available at PubChem URL: https://pubchem.ncbi.nlm.nih.gov (accessed in February 2022).

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
