# Peer review of "Bioactivity and Development of Small Non-Platinum Metal-Based Chemotherapeutics"

_pharmaceutics, 2022, doi:10.3390/pharmaceutics14050954_

Round 1

Reviewer 1 Report

In this Review Ferraro et al offer an overview on the status, applications and future development of inoraganic chemotherapeutics for cancer treatment. Specific attention is devoted to Ru, Pd, Rh, Ir and Au-based complexes.

Of course the amount of available literature is huge and it is hard to comprehensively outline the state of the art. This is why authors have competently focused their attention on the above metals. I believe this contribution fits with the scope of the SI and it is also a valuable reading for several people working in the field of bioinorganic chemistry, medicinal inorganic chemistry and more in general in cancer research.

According to this, I suggest publication provided the following points are addressed:

-line 26 abstract remove "complexes" after "iridium".

-discussing the development of cddp and its derivatives it would be nice to include a timeline reporting the dates for their discovery and approval.

-line 100-101: One of the most important paper on the oxaliplatin's mechanism of action id by Lippard et al (10.1038/nm.4291), it should be included. However, please consider that this mechanisms is not exclusive and the DNA still represent an important target even for oxaliplatin (see also 10.2174/1568026621666211004092333).

-In the introduction a figure reporting the most important inorganic drugs that entered clinical trials or that have been approved should be added 

-line 138: The sentence "Unlike well-known square-planar platinum(II) complexes, platinum(IV) complexes normally adopt octahedral geometries, producing chemically stable compounds that do not undergo ligand substitution" is misleading. Indeed, even Pt(IV) compounds may undergo ligand displacement. 

-In the Ru section it would be good to include a short discussion on the effects of Ru compounds on angiogenesis (see for instance 10.2174/1568026620666201126163436)

-In the Ir section it would be also interesting to add a brief discussion on the mechanism of action of Ir(I) compounds whose cytotoxicity might be associated to the Ir(I)-->Ir(III) oxidation 

-In the gold section I suggest authors to verify whether ATM is currently included in some clinical trials. I believe no trials with ATM are now active.

-In the conclusion section few sentences on the importance of mechanistically-driven drug discovery for improved inorganic drugs should be added.

Reviewer 2 Report

The topic of the proposed review is in a very important area, namely the advantages of the compounds of the platinum group metals and gold in application as drugs in chemotherapy against cancer. The shortcomings in the use of cisplatin and the world-accepted platinum-based drugs have been highlighted as the main reason for the search of new "non-classical" and "non-platinum" drugs. The review is focused on the huge, but still untapped potential that the small non-platinum metal-based compounds offer in the search and development of effective and safe drugs. The collected information as well as current research on small molecules' compounds of the metals Ruthenium, Palladium, Rhodium, Iridium and Gold designed as drugs is presented underlining the best formulas of the each metal. Special attention is paid to the available research on the mechanisms of bioactivity, biodistribution and biotransformation of selected compounds. The references are appropriately selected and provide useful information for researchers. This work would be improved if the characterization of the nature and specificities of individual metal ions were analyzed in more depth. From a chemical point of view, the properties of the individual ions such as oxidation state, redox potential, kinetic behavior, stability of chemical  bonds with different donor atoms of the drug’s ligands or biomolecules involved in the mechanisms is an important prerequisite for planning syntheses of new compounds with improved properties. In this regard, it would be better to discuss also the choice of ligands in the construction of already known drugs.

Reviewer 3 Report

The manuscript entitled "Bioactivity and Development of small non-platinum metal-based chemotherapeutics" provides the literature overview of applications of small non-platinum metal-based compounds in cancer chemotherapy. The manuscript is well organized but I do not think that the present review is within the central scope of the journal. The manuscript is mainly associated with inorganic chemistry and bioactivity of compounds. However, the formulation and drug targeting approaches for these compounds are missing. In my opinion, the manuscript is not suitable for publication in Pharmaceutics. 

Round 2

Reviewer 3 Report

Dear Authors

Thanks for your response to my comments. I have no objection to the quality and technical soundness of the work. My only concern was associated with the association of present work with the central scope of the journal. I did not see drug targeting, formulation, or drug delivery approaches in the present review. I request the honorable editor to evaluate the present review within the central scope of the journal and finalize the decision accordingly.